

# Diversity in root growth responses to moisture deficit in young faba bean (*Vicia faba* L.) plants

Kiflemariam Yehuala Belachew[1], Kerstin A. Nagel[2], Fabio Fiorani[2] and Frederick L. Stoddard[1]

[1] Department of Agricultural Sciences, Viikki Plant Science Centre, University of Helsinki, Helsinki, South Finland, Finland
[2] IBG-2: Plant Sciences, Forschungszentrum Jülich GmbH, Jülich, Germany

Corresponding author
Kiflemariam Yehuala Belachew,
kiflemariam.belachew@helsinki.fi

## ABSTRACT

**Background.** Soil moisture deficiency causes yield reduction and instability in faba bean (*Vicia faba* L.) production. The extent of sensitivity to drought stress varies across accessions originating from diverse moisture regimes of the world. Hence, we conducted successive greenhouse experiments in pots and rhizotrons to explore diversity in root responses to soil water deficit.

**Methods.** A set of 89 accessions from wet and dry growing regions of the world was defined according to the Focused Identification of Germplasm Strategy and screened in a perlite-sand medium under well watered conditions in a greenhouse experiment. Stomatal conductance, canopy temperature, chlorophyll concentration, and root and shoot dry weights were recorded during the fifth week of growth. Eight accessions representing the range of responses were selected for further investigation. Starting five days after germination, they were subjected to a root phenotyping experiment using the automated phenotyping platform GROWSCREEN-Rhizo. The rhizotrons were filled with peat-soil under well watered and water limited conditions. Root architectural traits were recorded five, 12, and 19 days after the treatment (DAT) began.

**Results.** In the germplasm survey, accessions from dry regions showed significantly higher values of chlorophyll concentration, shoot and root dry weights than those from wet regions. Root and shoot dry weight as well as seed weight, and chlorophyll concentration were positively correlated with each other. Accession DS70622 combined higher values of root and shoot dry weight than the rest. The experiment in GROWSCREEN-Rhizo showed large differences in root response to water deficit. The accession by treatment interactions in taproot and second order lateral root lengths were significant at 12 and 19 DAT, and the taproot length was reduced up to 57% by drought. The longest and deepest root systems under both treatment conditions were recorded by DS70622 and DS11320, and total root length of DS70622 was three times longer than that of WS99501, the shortest rooted accession. The maximum horizontal distribution of a root system and root surface coverage were positively correlated with taproot and total root lengths and root system depth. DS70622 and WS99501 combined maximum and minimum values of these traits, respectively. Thus, roots of DS70622 and DS11320, from dry regions, showed drought-avoidance characteristics whereas those of WS99501 and Mèlodie/2, from wet regions, showed the opposite.

**Discussion.** The combination of the germplasm survey and use of GROWSCREEN-Rhizo allowed exploring of adaptive traits and detection of root phenotypic markers

for potential drought avoidance. The greater root system depth and root surface coverage, exemplified by DS70622 and DS11320, can now be tested as new sources of drought tolerance.

## INTRODUCTION

Faba bean (*Vicia faba* L.) is an agronomically important crop for sustainable cropping systems (*De Visser, Schreuder & Stoddard, 2014*) and has value for both food and feed (*Crépon et al., 2010*). Drought poses a great challenge to the sustainable production of the crop (*Khan et al., 2010*). Most faba bean genotypes are sensitive to soil moisture loss and heat stress (*Loss, Siddique & Martin, 1996*), showing leaf wilting symptoms even at moderate soil water potential (*McDonald & Paulsen, 1997*). Yield losses and instability are the main problems of this crop in drought-affected areas (*Khan et al., 2010*). Nevertheless, faba bean shows drought adaptation potential in the field (*Reid, 1990*) and diversity exists in abiotic stress tolerance (*Khazaei et al., 2013*; *Belachew & Stoddard, 2017*). For example, line ILB938 has demonstrated drought tolerance in different experiments in controlled conditions (*Link et al., 1999*; *Khan et al., 2007*). *Khazaei et al. (2013)* studied the leaf morphophysiological traits of two sets of 201 faba bean accessions collected from dry and wet regions of the world, chosen according to the Focused Identification of Germplasm Strategy (FIGS), which is based on the concept that traits are the outward expressions of the environment in which the genotypes evolved. The ''dry set'' included accessions collected from sites where the annual rainfall was between 300 and 550 mm, whereas the ''wet set'' accessions were collected from sites receiving an annual rainfall of more than 800 mm (*Khazaei et al., 2013*). The results indicated the potential of FIGS in the search for target traits for drought stress adaptation, but its focus on leaf traits left root traits open for later study.

High-throughput screening and phenotyping of plants grown in pots allows controlled and uniform moisture stress, which is difficult to achieve under field conditions (*Tuberosa, 2012*). Screening of faba bean in well watered conditions provided initial information about leaf traits related to drought adaptation (*Khazaei et al., 2013*). Leaf chlorophyll content is a key trait determining the source capacity in affecting cumulative photosynthesis (*Tuberosa, 2012*) and in peanut (*Arachis hypogaea* L.), it is positively correlated with dry root biomass and used to discriminate accessions for drought stress (*Songsri et al., 2009*). Stomatal conductance and canopy temperature depression (CTD) are two methods to screen cool-season legumes for drought stress (*Stoddard et al., 2006*). Large stomatal response, which is the expression of sensitivity to soil moisture deficiency, is regarded as useful for long-term drought (*Munns et al., 2010*) and considered as a consistent indicator of growth rate response to stress. CTD, the difference in temperature between the canopy surface and the surrounding air, incorporates the

effects of multiple biochemical and morphophysiological features acting at the root, stomata and the plant canopy (*Tuberosa, 2012*). Accessions exhibiting cooler canopy temperature under drought stress avoid excessive dehydration through the use of more of the available moisture in the soil. Hence, CTD indicates plant water status in monitoring plant responses to water stresses (*Tarek et al., 2014*) and it is reported as the most responsive trait in faba bean accessions (*Khan et al., 2007*; *Khazaei, 2014*).

Together with shoot traits, identifying root phenotypic markers will help to understand the mechanisms by which they affect tolerance to drought. Root studies in legumes are relatively few and much less is known about roots than about shoots. When plants were grown in tall cylinders containing 1:1 Vertisol:Sand mixture (w/w), trait diversity for drought tolerance in chickpea (*Cicer arietinum* L.) was readily detected, including deeper rooting and greater biomass proportion in roots (*Kashiwagi et al., 2006*). Shovelomics and automated image phenotyping methods revealed genotypic variation in cowpea (*Vigna unguiculata* (L.) Walp.) root architecture, such as number of lateral roots and volume of soil enclosed by roots (*Burridge et al., 2017*). In a controlled environment, GROWSCREEN-Rhizo, a novel automated phenotyping robot, enables relatively high-throughput and non-invasive root phenotyping through characterization of root geometry (*Nagel et al., 2012*; *Gioia et al., 2015*; *Avramova et al., 2016*). With this tool, images are captured in real time and the functional and structural parts of the crop are quantified using image analysis software (*Nagel et al., 2009*; *Rascher et al., 2011*; *Nagel et al., 2012*).

For these reasons, we set out to investigate variation in root morphology of faba bean. The first hypothesis, tested with a germplasm survey, was that dry-zone germplasm would have more prolific root systems than wet-zone germplasm. The second hypothesis, tested with the phenotyping robot, was that dry-zone germplasm would maintain its root system growth better in drought than wet-zone germplasm would.

## MATERIAL AND METHODS

### Germplasm survey

The germplasm survey was conducted at the University of Helsinki's Viikki Campus greenhouse facility in a randomized complete block design (RCBD) with four replications. The four blocks were sown at seven-day intervals (owing to space limitation) and allowed to grow for 34 days and each block contained one pot of each accession. Throughout the experiment, the photoperiod was set at 14 h light and 10 h dark, and the temperature maintained at 22 °C during the day and 16 °C in the night.

The original set of 201 wet-adapted and 201 dry-adapted accessions (*Khazaei et al., 2013*) was reduced to 88 based on differences in canopy temperature depression measured in the glasshouse (*Khazaei et al., 2013*), country of origin and availability of seeds. Ten other accessions (seven from Ethiopia and three from Europe) were selected from the previous screening experiment for acid-soil and aluminium toxicity tolerance (*Belachew & Stoddard, 2017*). ILB938/2 and Mélodie/2 were included as they have been well studied previously (*Link et al., 1999*; *Khan et al., 2007*; *Khazaei et al., 2013*). Poor germination of 11 accessions reduced this set of 100 to 89, 44 of which were from the previous dry set, 38

**Table 1  List of experimental materials by country of origin and source.** GU is University of Göttingen; HARC is Holeta Agricultural Research Center, Ethiopia; ICARDA is International Center for Agricultural Research in the Dry Areas; INRA is French National Institute for Agricultural Research; Prefixes DS and WS indicate material originally allocated to the dry set and wet set (*Khazaei et al., 2013*).

| S.N. | Accessions | Country of origin | Source | S.N. | Accessions | Country of origin | Source | S.N. | Accessions | Country of origin | Source |
|---|---|---|---|---|---|---|---|---|---|---|---|
| 1 | Aurora | Sweden | Svalöf Weibull | 31 | DS137675 | Tajikistan | ICARDA | 61 | WS115134 | Nepal | ICARDA |
| 2 | Babylon | Netherlands | Nickerson Limagrain | 32 | DS13918 | Sudan | ICARDA | 62 | WS115177 | Nepal | ICARDA |
| 3 | DOSHA | Ethiopia | HARC | 33 | DS70622* | Syria | ICARDA | 63 | WS115182 | Nepal | ICARDA |
| 4 | DS11202* | Jordan | ICARDA | 34 | DS72271 | Morocco | ICARDA | 64 | WS115186 | Nepal | ICARDA |
| 5 | DS11207 | Syria | ICARDA | 35 | DS72309 | Syria | ICARDA | 65 | WS115352 | Nepal | ICARDA |
| 6 | DS112096 | Morocco | ICARDA | 36 | DS72310 | Syria | ICARDA | 66 | WS115430 | Nepal | ICARDA |
| 7 | DS11210 | Syria | ICARDA | 37 | DS72366 | Syria | ICARDA | 67 | WS11688 | Afghanistan | ICARDA |
| 8 | DS11236 | Iraq | ICARDA | 38 | DS72387 | Syria | ICARDA | 68 | WS117830 | China | ICARDA |
| 9 | DS11281 | Afghanistan | ICARDA | 39 | DS72396 | Syria | ICARDA | 69 | WS117841 | China | ICARDA |
| 10 | DS11286 | Iran | ICARDA | 40 | DS72455 | Syria | ICARDA | 70 | WS117849 | China | ICARDA |
| 11 | DS11294 | Spain | ICARDA | 41 | DS72493 | Syria | ICARDA | 71 | WS117853 | China | ICARDA |
| 12 | DS11317 | Macedonia | ICARDA | 42 | DS72523 | Syria | ICARDA | 72 | WS117855 | China | ICARDA |
| 13 | DS11320* | Macedonia | ICARDA | 43 | DS74370 | Oman | ICARDA | 73 | WS117857 | China | ICARDA |
| 14 | DS11437 | Turkey | ICARDA | 44 | DS74554 | Algeria | ICARDA | 74 | WS117864 | China | ICARDA |
| 15 | DS11480 | Lebanon | ICARDA | 45 | DS74573* | Russia | ICARDA | 75 | WS117868 | China | ICARDA |
| 16 | DS11561 | Algeria | ICARDA | 46 | DS99515 | Kyrgyzstan | ICARDA | 76 | WS12315 | Sweden | ICARDA |
| 17 | DS11591 | Tunisia | ICARDA | 47 | EH 06006-6* | Ethiopia | HARC | 77 | WS124242 | China | ICARDA |
| 18 | DS11689 | Afghanistan | ICARDA | 48 | Gebelcho | Ethiopia | HARC | 78 | WS13039 | Ethiopia | ICARDA |
| 19 | DS11701 | Afghanistan | ICARDA | 49 | GLA 1103 | Austria | Gleisdorf | 79 | WS130600 | Russia | ICARDA |
| 20 | DS11788 | Afghanistan | ICARDA | 50 | ILB938/2* | Ecuador | ICARDA /GU | 80 | WS130731 | Azerbaijan | ICARDA |
| 21 | DS11909 | Ethiopia | ICARDA | 51 | Kassa | Ethiopia | HARC | 81 | WS13107 | Greece | ICARDA |
| 22 | DS12257 | Syria | ICARDA | 52 | Mélodie/2* | France | INRA/GU | 82 | WS13185 | Turkey | ICARDA |
| 23 | DS124062 | Kazakhstan | ICARDA | 53 | Messay | Ethiopia | HARC | 83 | WS132238 | China | ICARDA |
| 24 | DS124138 | China | ICARDA | 54 | NC 58 | Ethiopia | HARC | 84 | WS132258 | China | ICARDA |
| 25 | DS124353 | Greece | ICARDA | 55 | Tesfa | Ethiopia | HARC | 85 | WS132266 | China | ICARDA |
| 26 | DS13042 | Italy | ICARDA | 56 | WS11309 | Poland | ICARDA | 86 | WS132274 | China | ICARDA |
| 27 | DS131708 | Tajikistan | ICARDA | 57 | WS11313 | Ethiopia | ICARDA | 87 | WS99379 | Portugal | ICARDA |
| 28 | DS13463 | Cyprus | ICARDA | 58 | WS11344 | Russia | ICARDA | 88 | WS99465 | China | ICARDA |
| 29 | DS13473 | Cyprus | ICARDA | 59 | WS114476 | Bangladesh | ICARDA | 89 | WS99501* | China | ICARDA |
| 30 | DS13481 | Cyprus | ICARDA | 60 | WS114576 | Bangladesh | ICARDA | | | | |

**Notes.**

*indicates accessions used in the subsequent root phenotyping experiment.

from the wet set and 7 from Ethiopian highlands that conform to the criteria of the wet set (Table 1). Since seed quantities were limited, seed size was evaluated as one-tenth of 10-seed weight rather than hundred- or thousand-seed weight.

The experiment was designed to maximize expression of potential root mass by providing plentiful moisture and nutrients. The pots were 3 L in capacity, 20 cm deep and 15 cm diameter with four drainage holes of 2 cm diameter. The bottom of each pot was covered

with a thin membrane sheet and then the pots were filled with 0.2 L of sand at the bottom, 2.6 L of perlite, and 0.2 L of sand on the top. Two seeds per pot were sown and after five days, the weaker seedling was removed, leaving the stronger seedling to grow. Nutrient solution was applied at 200 mL automatically every other day from sowing to harvesting for 34 days to keep the medium at field capacity. Pests (thrips) were controlled biologically with parasitic wasps. The nutrient solution was 1 g/L of Superex Peat (Kekkilä Oy, Vantaa, Finland) supplemented with 2 mmol/L $CaCl_2$, as previously described (*Belachew & Stoddard, 2017*).

At BBCH stage 39 (*Meier, 2001*), when there were approximately 9 visibly extended internodes, 30-34 days after sowing (DAS), the following measurements were taken. Stomatal conductance was measured using a Leaf Porometer (Decagon Devices, Inc, Pullman, WA, USA) once per plant. Leaf surface temperature was measured using a FLUKE Model 574 Precision Infrared Thermometer (Fluke Corporation, Everett, WA, USA), chlorophyll concentration was measured as leaf SPAD values from two leaves per plant and the average of the two was recorded using SPAD-502 (Minolta Camera Co, Ltd, Japan). Measurements were taken between 11:00 and 13:00 local time. Plants were harvested at 34 DAS. Shoots were removed above the collar region and roots were carefully removed from the perlite. Both parts were dried in a drying oven at 70 °C for 48 h. Root and shoot dry weight were measured to the nearest 0.01 g and root to shoot dry weight ratio was calculated by dividing the root weight by the corresponding shoot weight. Root mass fraction was calculated as root dry mass divided by total plant dry biomass.

## Root phenotyping experiment

The experiment was conducted at Jülich Plant Phenotyping Center (JPPC) (http://www.jppc.de), Forschungszentrum Jülich GmbH, Germany from 23 January to 20 February 2017.

Eight accessions were chosen (Table 1) from the germplasm survey according to their performance in stomatal conductance, canopy temperature, chlorophyll concentration, root and shoot dry weights and root mass fraction values. Accessions showing high values of stomatal conductance and leaf surface temperature were considered as potentially drought susceptible, whereas those showing high chlorophyll concentration, root dry weight, root to shoot dry weight ratio, and root mass fraction, along with low values of stomatal conductance and leaf surface temperature were considered as potentially drought tolerant.

The experiment was conducted in the automated root and shoot phenotyping platform GROWSCREEN-Rhizo using rhizotrons with a size of 90 × 70 × 5 cm (*Nagel et al., 2012*). The growth medium used was GRAB-ERDE, a dark peat-based substrate (Plantaflor Humus Verkaufs-GmbH, Germany). A total of 2,400 L peat was first machine broken and then passed through a 0.8 cm sieve. The initial moisture content of the peat-soil was 66.3% measured using Electronic Moisture Analyzer (version 1.1, 03/2013, KERN and Sohn GmbH, Germany). Of this, 1,600 L was air dried to 40% moisture content, when it had a water potential of 0.006 MPa according to the water retention curve analysis conducted by the Institute of Plant Nutrition and Soil Science, University of Kiel, Germany.

Nutrient content and other physical and chemical properties of the growth medium were analyzed by LUFA NRW Laboratory, Germany. Dry matter content was 35%, wet bulk density 450 g/L, dry bulk density 158 g/L, pH 5.8, EC 733 μS/cm, KCl in $H_2O$ 1.76 g/L, KCl in $CaSO_4$ 0.45 g/L, total nitrogen 27 mg/L in $CaCl_2$/DPTA-Extract (CAT, where DPTA is diethylenetriamine-pentaacetic acid), $NH_4^+$-N 4 mg/L in CAT, $NO_3^-$-N 23 mg/L in CAT, $P_2O_5$ 22 mg/L in CAT, $K_2O$ 178 mg/L in CAT, Mg 125 mg/L in CAT, and Mn 11 mg/L in CAT.

Water-limited treatment boxes were filled with air dried peat-soil, whereas well watered treatment boxes were filled without drying. Each rhizotron contained approximately 21 L of growth medium. Filling was done in three steps of 7 L peat-soil each followed by regular pressing, to make the compaction of the medium as uniform as possible among boxes. The boxes were then fixed in the robotic system in the greenhouse and tilted at 43° from vertical.

## Research design

The experiment was arranged in a split-plot design, with four replicate blocks, two treatments (well watered and water limited) as the main plots and eight accessions as subplots.

## Planting and treatment management

The experiment was conducted for 28 days, from seed soaking to plant harvesting, during the vegetative stage of plant growth. Seeds of uniform size were selected from all 8 accessions, washed three times, surface sterilized with 1% NaClO (sodium hypochlorite) (w/v) for 5 min and rinsed 3 times with running tap water. The seeds were soaked in tap water for 24 h, transferred to three layers of moist filter paper in 14 cm diameter Petri dishes (14 seeds/dish) as described in *Belachew & Stoddard (2017)*, and incubated for 96 h at 22 °C in the dark. The seedlings showing uniform root growth were selected and transferred into the rhizotrons. Initially, for establishment, each seedling in well watered treatment received 200 ml water in the automatic irrigation system and those in water limited treatment received 50 ml of water to their roots manually. Following this, the well watered plants were given 100 ml of water every 12 h until the end of the treatment period. In the water-limited treatment, plants received the second 50 ml of water four days after transplanting and thereafter received no more water. The average peat-soil temperature was 22.6 °C, air humidity 58% and air temperature 20.9 °C. The photoperiod was 15 h light and 9 h dark.

## Data collected

Root images were automatically taken every day except on Saturday and Sunday from 30 January to 20 February 2017. Images taken 5 days after treatment (DAT), 12 DAT, and 19 DAT were analyzed using the PaintRHIZO software package and dimensions were converted to SI units using 55.53 pixel = 1 cm. The following root distribution and individual root traits were computed (*Nagel et al., 2009*; *Nagel et al., 2012*):

- taproot length (cm);
- first and second order lateral roots length (cm);

- total root length (cm);
- root system depth (cm), which represents the maximum vertical distribution of the root system;
- root system width (cm), which represents maximum horizontal distribution of the root system; and
- convex hull area (cm$^2$), which measures the surface area along the transparent plate of the rhizotrons covered by a root system.

To evaluate how much of the whole root system was visible at the transparent plate of the rhizotrons, we measured total root length destructively, using accession DS70622 as a test-case because it had the largest root system in both irrigation treatments. The roots were carefully removed from the potting medium 19 days after the treatment, washed, and preserved in ethanol solution until analysis. One week later, each root system was thoroughly washed, cut into manageable lengths and spread in water on the WinRhizo scanner.

## Data analysis

Root images obtained with GROWSCREEN-Rhizo and manual root scanner EPSON A3 Transparency Unit (Model EU-88, Japan) were analyzed using PaintRHIZO and WinRHIZO, respectively, following the methods developed by *Mühlich et al. (2008)*, *Nagel et al. (2009)* and *Nagel et al. (2012)*.

Quantitative data from the survey and the phenotyping experiment were subjected to analysis of variance using SPSS version 22.0 (IBM Inc., Chicago, IL, USA). Frequency distributions of the survey data showed that most measures gave acceptable fits to the normal distribution. Stomatal conductance and root-to-shoot dry weight ratio both showed excess kurtosis values above 4, and their skewness values were 0.9 and 1.5, respectively, owing to a single outlying accession in each case. Treatment means were separated by Duncan's Alpha (5%). The difference between the group means of the dry-adapted and wet-adapted sets in the germplasm survey was tested using an independent-samples $t$-test. In the phenotyping experiment, the two-way ANOVA tested the main effect of treatment, the main effect of accession, and the treatment by accession interaction effect on each sampling date (5, 12, and 19 DAT) separately. A $t$-test was used to compare the difference in root visibility of DS70622 between watering treatments.

## RESULTS

### Germplasm survey

There were significant differences between accessions in stomatal conductance, chlorophyll concentration, root and shoot dry weight and root mass fraction values ($P < 0.001$), root to shoot dry weight ratio ($P < 0.01$) and leaf surface temperature ($P < 0.05$). Stomatal conductance ranged 6-fold, shoot dry weight 12-fold, root dry weight 7-fold, root mass fraction 2-fold, seed size 18-fold and leaf surface temperature by 3.1 °C (Table 2). Accessions originating from dry growing regions of the world showed significantly higher chlorophyll concentration ($P < 0.001$), shoot ($P < 0.01$) and root ($P < 0.001$) dry weights than those from wet regions (Table 2 and Table S1).
**Table 2  Mean values of shoot and root measurements of 44 faba bean accessions from dry zones and 45 from wet zones. Seed weight data were unreplicated.**

| Data | Stomatal conductance (mmol $H_2O/m^2/s$) | Leaf surface temperature (°C) | Chlorophyll concentration (SPAD value) | Shoot dry weight (g) | Root dry weight (g) | Root to shoot dry weight ratio | Root mass fraction | Seeds weight (g) |
|---|---|---|---|---|---|---|---|---|
| Minimum | 109 | 20.7 | 24.1 | 0.27 | 0.21 | 0.31 | 0.24 | 0.12 |
| Mean | 316 | 22.2 | 33.1 | 1.82 | 0.80 | 0.47 | 0.32 | 0.99 |
| Maximum | 752 | 24.8 | 41.0 | 3.49 | 1.59 | 0.94 | 0.50 | 2.11 |
| Mélodie/2 | 252 | 23.3 | 38.7 | 0.74 | 0.38 | 0.51 | 0.32 | 0.5 |
| ILB 938/2 | 383 | 21.7 | 35.8 | 1.92 | 0.84 | 0.44 | 0.30 | 1.04 |
| SE | 65 | 0.7 | 1.6 | 0.27 | 0.12 | 0.13 | 0.04 | |
| LSD (5%) | 182 | 2.0 | 4.4 | 0.75 | 0.35 | 0.37 | 0.10 | |
| **P-value (accessions)** | *** | * | *** | *** | *** | ** | *** | |
| Mean dry set | 323 | 22.1 | 34.4 | 2.02 | 0.91 | 0.46 | 0.33 | 1.15 |
| Mean wet set | 308 | 22.2 | 31.7 | 1.63 | 0.71 | 0.47 | 0.32 | 0.68 |
| **P-value (sets)** | ns | ns | *** | ** | *** | ns | ns | |

Notes.
*$p < 0.05$.
**$p < 0.01$.
***$p < 0.001$.
SE, standard error; LSD, least significant difference.

**Table 3  Pearson correlations of shoot and root data of 89 faba bean accessions.**

| | Stomatal conductance | Leaf surface temperature | Chlorophyll concentration | Shoot dry weight | Root dry weight | Root to shoot dry weight ratio | Root mass fraction |
|---|---|---|---|---|---|---|---|
| Canopy temperature | −0.14 | | | | | | |
| Chlorophyll concentration | −0.23* | −0.04 | | | | | |
| Shoot dry weight | −0.11 | 0.05 | 0.12 | | | | |
| Root dry weight | −0.01 | −0.05 | 0.23* | 0.89** | | | |
| Root to shoot dry weight ratio | 0.08 | −0.05 | 0.08 | −0.60** | −0.24* | | |
| Root mass fraction | 0.07 | −0.03 | 0.09 | −0.56** | −0.21* | 0.90** | |
| Seed weight | −0.78** | 0.001 | 0.36** | 0.61** | 0.58** | −0.34** | −0.23* |

Notes.
*$P < 0.05$ (2-tailed).
**$P < 0.01$ (2-tailed).

Chlorophyll concentration showed a weak but significant negative correlation with stomatal conductance and a similarly weak but positive one with root dry weight (Table 3). Root and shoot dry weight were positively correlated with each other (Fig. 1) and with seed weight (Table 3). Root mass fraction was negatively correlated with seed weight.

The five accessions with the greatest root and shoot weights were from the dry set and the five with the lowest were from the wet set (Fig. 1). The accessions with the two greatest total dry weights were DS70622 (5.1 g) and DS74573 (4.8 g) (Fig. 1). Accession DS11320 was an outlier with the highest value of stomatal conductance, along with a one of the highest values of root dry weight. Accession WS114476 was an outlier with the highest value
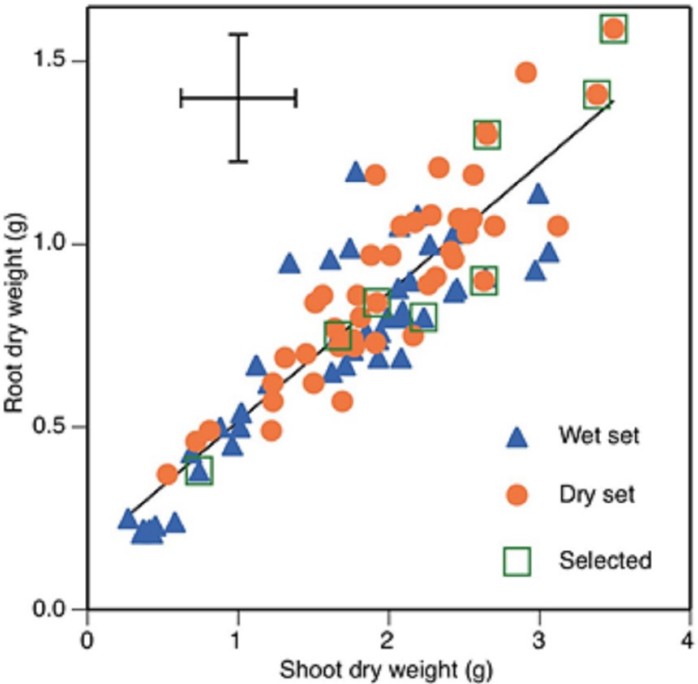

**Figure 1** **Root and shoot dry weights of 89 faba bean accessions, 45 from wet zones and 44 from dry zones.** Selected accession, from lower left to upper right, are Mélodie/2, WS99501, ILB938/2, EH06006-6, DS11202, DS11320, DS74573 and DS70622. Error bars show least significant difference. Regression line shows root dry weight = 0.353 * shoot dry weight +0.162, $r^2 = 0.787$.

**Table 4** **Faba bean accessions chosen for root phenotyping experiment and the bases of selection in the screening experiment.**

| Chosen accessions | Selection criteria |
| --- | --- |
| DS11202 | High leaf surface temperature, low chlorophyll concentration, low root mass fraction and root to shoot dry weight ratio |
| DS11320 | Low leaf surface temperature, high root and shoot dry weights |
| DS70622 | Low leaf surface temperature, high root and shoot dry weights |
| DS74573 | High shoot and root dry weight |
| EH 06006-6 | High leaf surface temperature, low chlorophyll concentration, low root mass fraction and low root to shoot dry weight ratio |
| ILB 938/2 | Benchmark from previous research for drought tolerance |
| Melodie/2 | Benchmark from previous research for efficient use of water |
| WS99501 | High stomatal conductance, high leaf surface temperature, low root weight, low root to shoot ratio and low root mass fraction |

of root-to-shoot dry weight ratio, but this was combined with very low total dry weight production. Eight accessions (Table 4) were chosen for the root phenotyping experiment.

## Root phenotyping

The water-limited treatment was sufficiently strong to reduce the lengths of all three classes of root (taproot, lateral and second order lateral roots) at all three time points (5, 12

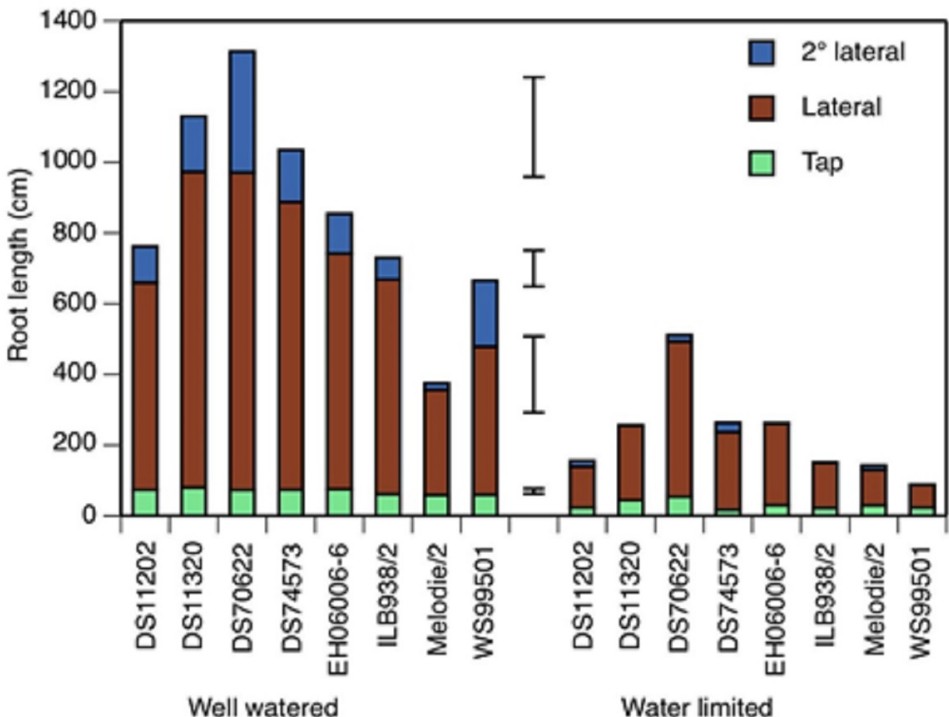

**Figure 2 Tap root, lateral and second order lateral root lengths of 8 accessions of faba bean in two water treatments at 19 days after initiation of treatment.** Total root length is the sum of the three classes. Error bars show least significant differences of, bottom to top, taproot, lateral, second order lateral, and total root length.

and 19 days after treatment started (DAT)) below the values found in the well watered treatment (Fig. 2 and Table S2). The main effect of accession on all three root lengths was also significant at all time points. The treatment × accession effect was significant for taproot and second order lateral root lengths at 12 and 19 DAT, but not for lateral root lengths, in which the standard error was large. Lateral roots made the largest contribution to total root length at 19 DAT (Fig. 2), 76% in well watered and 79% in water limited, a non-significant difference.

At 19 DAT, accession DS70622 had the longest lateral roots in both treatments, the longest second order lateral roots in the well watered treatment, the greatest total root length in both treatments, and the smallest difference in taproot and lateral root growth between treatments (Fig. 2). DS11320 had the longest tap root, the second-longest laterals and the second-longest total root length in the well watered treatment. EH06006-6 had the second-longest taproots in the well watered treatment. Mélodie/2 had the shortest taproot, lateral and second order lateral roots in the well watered treatment, whereas in the water-limited treatment, DS74573 had the shortest tap root, and WS99501 had the shortest laterals, second order laterals and total root length (Fig. 2).

**Table 5  Mean root system depth and convex hull area of 8 faba bean accessions at 19 DAT, $n = 4$.**

| Accessions | Root system depth (cm) | | Convex hull area (cm²) | |
|---|---|---|---|---|
| | Well watered | Water limited | Well watered | Water limited |
| DS11202 | 74 | 29 | 2,061 | 410 |
| DS11320 | 78 | 46 | 2,491 | 927 |
| DS70622 | 76 | 53 | 2,515 | 1,047 |
| DS74573 | 76 | 35 | 2,369 | 663 |
| EH 06006-6 | 78 | 34 | 2,793 | 679 |
| ILB938/2 | 65 | 31 | 1,938 | 397 |
| Melodie/2 | 65 | 32 | 1,471 | 476 |
| WS99501 | 61 | 27 | 1,592 | 348 |
| SE | 3 | | 162 | |
| LSD (5%) | 8 | | 462 | |
| Overall | 72 | 36 | 2,154 | 618 |
| SE | 1 | | 81 | |
| LSD (5%) | 4 | | 231 | |
| *P*-value | | | | |
| Treatment | *** | | *** | |
| Accession | *** | | *** | |
| Treatment × Accession | ns | | ns | |

**Notes.**
***$p < 0.001$.
ns, not significant; SE, standard error; LSD, least significant difference; DAT, days after treatment given.

In three of the eight accessions, second order lateral roots were not visible at 5 DAT (Table S2). In the water-limited condition, only two of the accessions showed second order lateral roots at 12 DAT, but at 19 DAT, all of the test materials had these roots.

At 19 DAT, genotypic mean total root length and genotypic mean root system depth were positively correlated ($r = 0.86$, $n = 8$, $P < 0.01$), as were taproot length and total root length ($r = 0.82$, $n = 8$, $P < 0.05$).

At the end of the treatment period, the genotypic mean total root length of DS70622 was 3 times longer than those of Mélodie/2 and WS99501. Accessions DS11320 and DS70622 showed the two deepest root systems consistently at all 3 time points and WS99501 had the shallowest root system (Table 5 and Table S3).

On average, the total root length and root system depth recorded under well watered condition was twice that in the water-limited condition. Droughted roots continued to grow throughout the experiment, but more slowly than in well watered conditions, such that the total root length of the droughted treatment was 50%, 41% and 27% of non-droughted at 5, 12, and 19 DAT, respectively (Fig. 2 and Table S3). Similarly, root system depth was reduced by 40%, 46%, and 50%, respectively, at these three time points (Table 5 and Table S3).

Comparison of total root length records obtained from PaintRHIZO and WinRHIZO image analysis software of accession DS70622 indicated that roots in rhizotrons were 32.4% visible. The difference in visibility between the two treatments, 25.5% in the well watered

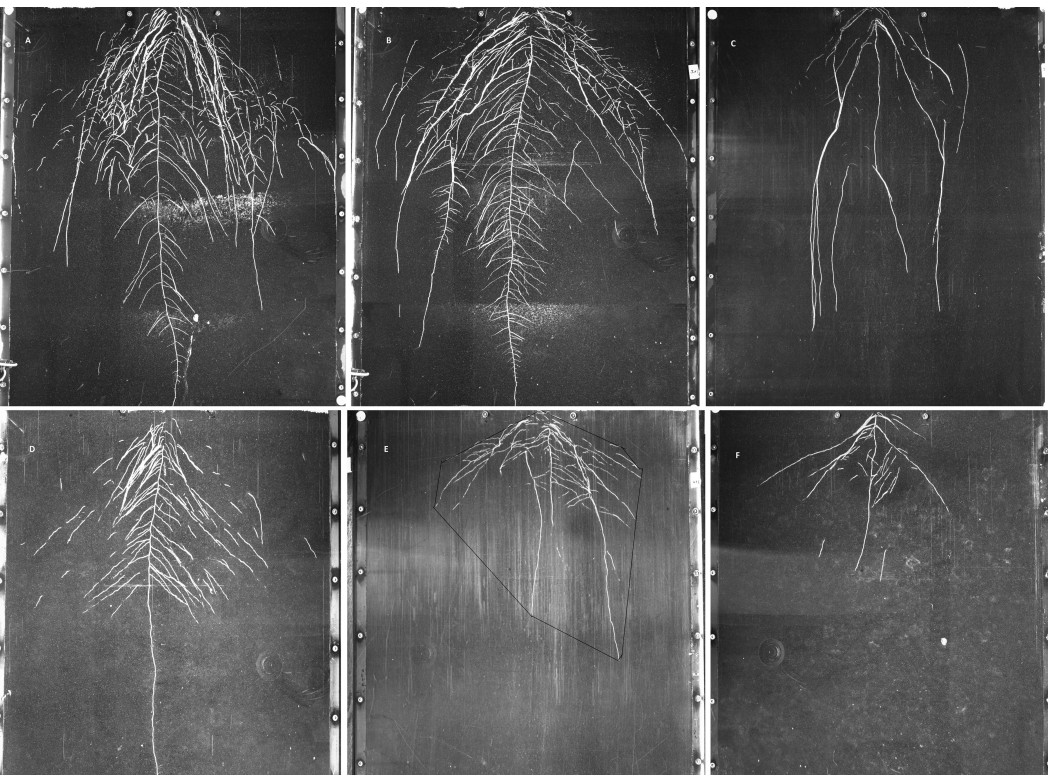

**Figure 3** **Examples of GROWSCREEN-Rhizo root images at 19 DAT.** (A–C): DS70622, DS74573, Mèlodie/2, respectively, in well watered treatment; (D–F) in the same order of accessions in water limited treatment. The outlined area in image E shows the convex hull area. Each image shows the full 70 cm width of the RhizoBox.

condition and 39.3% in the water-limited condition, was not statistically significant by a $t$-test.

Root system width differed between treatments, being 46 cm in the well watered condition and 28 cm in the water-limited treatment, but there was no significant difference between accessions.

Convex hull area showed large differences between treatments and between accessions (Fig. 3), but the interaction was not significant (Table 5). Treatment differences in convex hull area increased across the 3 time points (Table 5 and Table S3). Plants grown in the well watered condition showed about 3 times more convex hull area than plants grown in the water-limited condition. Maximum convex hull area was shown in accession DS70622, closely followed by EH06006-6 and DS11320, while WS99501 and Mélodie/2 had the two minimum values (Table 5). Root system width and convex hull area (genotypic means) were positively correlated ($r = 0.97$, $n = 8$, $P < 0.01$), and both traits were positively correlated with taproot length, total root length, and root system depth ($r = 0.82$ to $0.89$, $n = 8$, $P < 0.01$, $P < 0.05$).

## DISCUSSION

The germplasm survey showed that there was wide variation in morphological root traits of faba bean and that they were correlated with shoot traits, but that there were important outliers from that correlation. In the root phenotyping experiment, the water deficit was sufficiently harsh that it affected the length and width of all root systems, but there were large differences among accessions. Accession DS70622 had a larger root system than the benchmark drought-tolerant accession, ILB938/2, so it may be a potential source of genes for drought avoidance by improved access to soil water. These results are discussed below.

Accessions from dry regions of the world showed higher chlorophyll concentration, and root and shoot dry weight than those from wet regions in the survey. Increased chlorophyll concentration and SPAD value were observed due to drought in peanut genotypes and chlorophyll stability was reported to be an indicator of drought tolerance in that species (*Arunyanark et al., 2008*). High SPAD values and high root dry weight were positively correlated in lentil (*Lens culinaris* Medik) in response to drought (*Kumar et al., 2012*) and the decrease in chlorophyll content in drought tolerant genotypes of barley (*Hordeum vulgare* L.) was much less than in drought susceptible ones (*Li et al., 2006*). The correlation between growth of different plant parts is expected, and it leads to relatively consistent root:shoot ratio or root mass fraction (RMF). The outliers from the correlation are interesting as sources of potential breeding traits. In the present survey, root mass fraction ranged relatively widely, from 0.24 to 0.50, at 34 DAS. In a set of 211 chickpea accessions, RMF ranged from 0.38 to 0.53 at 35 DAS (*Kashiwagi et al., 2005*). The generally higher value of chickpea RMF may relate to its acknowledged greater drought tolerance. The outliers above regression line (Fig. 1) in the current set of faba bean were mostly in accessions from the "wet set", indicating that there may be useful sources of drought tolerance among this material. Our recalculations of RMF values from literature show higher values in each paper from drought-tolerant lines than from drought-susceptible ones: 0.57 to 0.66 in 133 recombinant inbred lines of lentil (*Idrissi et al., 2015*), 0.44 to 0.47 in 40 genotypes of lentil (*Sarker, Erskine & Singh, 2005*), and 0.20 to 0.25 in cowpea (*Vigna unguiculata* (L.) Walp.) genotypes (*Matsui & Singh, 2003*).

The substantial reduction in root length early in the phenotyping experiment emphasizes the importance of establishing faba beans with adequate moisture, particularly in agricultural regions subject to water deficit (*Loss, Siddique & Martin, 1996*). The reduction in root length was highly variable among accessions, being as high as 77% in DS74573 and as low as 30% in DS70622 (Fig. 2) at 19 DAT. This variation was shown to be significant in the accession by treatment interaction beginning from 12 DAT. The taproot of DS70622 in the water-limited condition was nearly 6× and 3× longer than those of WS99501 and Mèlodie/2, respectively. Similarly, drought-tolerant cultivars of common bean showed deeper roots than the sensitive ones (*Sponchiado et al., 1989*). Increased fine root length density and fine root dry weight was reported in white lupin (*Lupinus albus* L.) in response to water deficit (*Rodrigus, Pacheco & Chaves, 1995*). Lentil genotypes with longer roots and greater root dry weight were reported to tolerate terminal drought better than those with shorter roots and lower root weight (*Kumar et al., 2012*). Deep-rooted pulses can

benefit from stored water in times of drought more readily than shallow-rooted ones (*French & White, 2005*).

Root phenotyping technology provides new opportunities for assessing the effect of stress on different classes of root. Drought limited the length of laterals and second order lateral roots beginning from the onset of the treatment period. In sorghum (*Sorghum bicolor* Moench), the production of seminal root laterals was hindered by drought at the onset of the treatment and nodal roots produced few laterals only after some time (*Pardales & Kono, 1990*). Chickpea produced longer laterals when sown with sufficient moisture than when droughted (*Krishnamurthy, Johansen & Ito, 1994*). Mélodie/2 and WS99501 showed the greatest detrimental effect of drought already at 5 DAT and continued in that way for the rest of the experiment (Fig. 2 & Table S2). Even DS70622, the most prolifically rooting accession, did not show second order lateral roots in the water-limited condition until at least 12 DAT. Though the formation was first noted late, at 19 DAT, this accession was found to have the second longest second order lateral roots next to DS74573.

There were positive correlations between root area coverage (root system width and convex hull area) and root depth (tap root and total root lengths, and root system depth) measurements, indicating that faba beans expand their root system in depth and breadth in a more or less balanced way. However, though convex hull area showed large differences between treatments and among accessions, there was no accession by treatment interaction suggesting a strong genetic effect. Drought-tolerant chickpea genotypes showed adaptive root distribution, with a higher root length density at deeper soil layers during a severe drought year (*Kashiwagi et al., 2006*), whereas roots of this species remained near the surface in moist conditions (*Benjamin & Nielsen, 2006*). This plasticity is especially important for the crop to avoid both terminal drought (*Kashiwagi et al., 2006*; *Gaur, Krishnamurthy & Kashiwagi, 2008*) as well as transient drought. Peanut genotypes with a large root system showed high water use efficiency under drought condition (*Songsri et al., 2009*). Prolific and deep root systems have been shown in drought-avoiding accessions of chickpea (*Kashiwagi et al., 2005*), cowpea (*Matsui & Singh, 2003*), field pea (*Pisum sativum* L.) and soybean (*Glycine max* (L.) Merr.) (*Benjamin & Nielsen, 2006*). Hence, accessions with a larger root system probably avoid drought through increased access to water in the soil by increased tap root length as well as overall root system depth and width.

In the germplasm survey, the benchmark accessions Mélodie/2 and ILB938/2 showed low stomatal conductance, high chlorophyll concentration, and low shoot and root dry weight as compared to the rest. This was in agreement with the findings of *Khazaei et al. (2013)* in which Mélodie/2 and ILB 938/2 were reported to express efficient use of water and water use efficiency, respectively. In the root phenotyping experiment, however, the two accessions performed well below other accessions such as DS70622 and DS11320. This contradiction might be due to the initiation of the treatment at a much earlier stage of growth, which is in agreement with the finding that the root distribution of peanut genotypes at 37 and 67 days after sowing did not adequately predict the effects of drought, and best prediction being obtained at 97 days after sowing (*Songsri et al., 2008*). There are many ways in which plants respond to water deficit (*Pereira & Chaves, 1993*). Those from dry areas may show tolerance by increased root system depth and cavitation resistance

(*Hacke, Sperry & Pittermann, 2000*), root growth at the expense of above-ground parts (*Husain et al., 1990*; *Reid, 1990*), osmotic regulation and solute buildup, and expression of aquaporins (*Lian et al., 2004*; *Galmés et al., 2007*). Crop plants that tolerate drought through the biosynthesis of abscisic acid (ABA) may also show reduced water use and low biomass production because of low leaf growth, low stomatal conductance (*Galmés et al., 2007*) and hence low photosynthesis even in wet growing conditions (*Tardieu, 2003*). ILB938/2 follows this model. Other plant internal changes can regulate the opening of stomata as well (*Galmés et al., 2007*).

## CONCLUSIONS

The GROWSCREEN-Rhizo phenotyping platform allowed detection of useful differences in root responses to water deficit. In both the survey and the rhizotron experiments, the shoot and root traits varied widely among accessions, and these traits were positively correlated among each other. In both cases, higher values of morpho-physiological shoot and root measurements were recorded from accessions originating from the drier growing regions of the world, confirming the significance of FIGS to identify drought-adaptive traits.

The growth of the root system of faba bean in depth and width followed a balanced pattern, a strategy of wider and deeper soil exploration for water. Accession DS70622 produced the greatest root mass in the survey and phenotyping treatments, and maintained its root mass and convex hull area under stress. In the water-limited treatment, accession DS11320 produced considerably less root mass than DS70622, but combined this with a high convex hull area, and in the survey it had by far the highest stomatal conductance, suggesting that it was efficient at finding water. Thus, these two accessions can be new sources of root traits for future breeding of drought tolerant cultivars.

## ACKNOWLEDGEMENTS

We acknowledge Sebastien Carpentier, Astrid Junker, and Andreas Voloudakis for facilitating the STSM application with COST Action FA 1306. We thank Markku Tykkyläinen, Jouko Närhi and Sanna Peltola for their greenhouse assistance in the screening experiment in Helsinki, Anna Galinski, Henning Lenz, Bernd Kastenholz, Beate Uhlig, Carmen Müller, Jonas Lentz, and Ann-Katrin Kleinert for their assistance in the laboratory and greenhouse activities in Jülich, and Hendrik K. Poorter for his valuable comments during the conduct of the research. We also thank Jülich Plant Phenotyping Center (JPPC), IBG-2: Plant Sciences for hosting the root phenotyping research in their facility.

### Funding

This work was supported by the Finnish Cultural Foundation (Business ID 0116947-3) and COST Action FA1306 (COST-STSM-FA1306-35913). There was no additional external

funding received for this study. The funders had no role in study design, data collection and analysis, decision to publish, or preparation of the manuscript.

## Grant Disclosures
The following grant information was disclosed by the authors:
Finnish Cultural Foundation: 0116947-3.
COST Action: FA1306.

## Competing Interests
Kerstin A. Nagel and Fabic Ficrani are employees of Forschungszentrum Jülich GmbH.

## Author Contributions
- Kiflemariam Yehuala Belachew conceived and designed the experiments, performed the experiments, analyzed the data, prepared figures and/or tables, authored or reviewed drafts of the paper, approved the final draft.
- Kerstin A. Nagel contributed reagents/materials/analysis tools, prepared figures and/or tables, authored or reviewed drafts of the paper, approved the final draft.
- Fabio Fiorani prepared figures and/or tables, authored or reviewed drafts of the paper, approved the final draft.
- Frederick L. Stoddard conceived and designed the experiments, analyzed the data, contributed reagents/materials/analysis tools, prepared figures and/or tables, authored or reviewed drafts of the paper, approved the final draft.

## Data Availability
The raw data is included in Tables S1–S6.

## Supplemental Information
Supplemental information for this article can be found online at http://dx.doi.org/10.7717/peerj.4401#supplemental-information.

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
