# Peer review of "Diversity in root growth responses to moisture deficit in young faba bean (Vicia faba L.) plants"

_PeerJ, doi:10.7717/peerj.4401_

## Round 0.1 · original submission · Major Revisions

Your manuscript has been seen by two qualified reviewers. Based on their detailed assessment and my own, I feel this manuscript could be suitable for publication in PeerJ following a number of major revisions. In particular, the authors need to provide more details on the experiential approaches, rational for sampling and experimental design, and improve the overall clarity of the manuscript.

·

Basic reporting

The English is satisfactory and the professional article structure was followed.

Experimental design

The topic of the research is crucial and fit within the scope of the journal.

• The main concern regarding this experiment is the method used for selecting the potential drought susceptible and resistant accessions. The authors screened a panel of 89 accessions in a well-watered condition and based on the physiological responses like stomatal conductance, leaf temperature, chlorophyll content root/shoot dry matter, eight accessions were selected for further root architecture phenotyping. Although these traits are important, they do not necessarily translate to the ultimate drought resistant phenotype. The drought resistant cultivars can use different strategies according to Levitt (1980) and one of them is increasing the root architecture (mainly observed in water spenders as a avoidance strategy). These complexities of drought tolerance should be acknowledged in the experiments addressing the drought stress. I would also recommend to choose the drought resistant/susceptible accessions directly by evaluating the panel (89 lines) under drought condition and compare the results with the well-watered treatment.

Minor comment:
• In Line 101 please indicate the type of checks: tolerant or susceptible checks?!
• Line 102: the final number of accessions were 89 so please simply mention the origin of these 89 lines. What portions is from wet/dry origin? How many from Ethiopia and Europe?..
• Please mention the growing condition of the 89 accessions. Did you have different treatments (control vs drought)?!
• Line 117: please use “leaf surface temperature” instead of “canopy temperature” for this sentence and the whole manuscript.
• Line 117: Instead of “days after sowing” please indicate the developmental stage of the plants. It is difficult to interpret the developmental stage from days after sowing since growing rate depends on several environmental and genetic factors.
• Line 209: Please put a space in “Chlorophyll concentration was”
• The structure of Table 5 is confusing. For instance “Well-watered”/”water limited” were repeated in both columns and rows. Why not to replace them with “overall” in the row?!
• Please explain the abbreviations for each table like SE and LSD.
• Line 290: Please replace “genotype by environment interaction” with “genotype by treatment interaction”. Environment is usually refer to source of variation in the experiment with random effects.
• In table 2 and 4 please indicate the values of two checks. Also in Figure 1 highlight the checks.

Validity of the findings

• It is crucial to assess the correlation between greenhouse results and the phenotypes in the field. Root architecture is highly dynamic and affected by the environmental factors. Using the root architecture parameters as markers in the selection process by breeders is applicable if it is confirmed in the field.

Reviewer 2 ·

Basic reporting

I believe this experiment was a good use of available resources with the goal to use an advanced root phenotyping technique to identify traits for comparing previously described as being from “dry” and “wet” regions of the world.

1. Basic Reporting
Overall, this manuscript is not clear in many areas of the paper and needs to be improved structurally and grammatically. Often, terms are used without being defined or with a reference which makes it difficult to read. For example, not
Line 15: 89 accessions from wet and dry growing regions of the world
It’s not clear why you chose these accessions, what you mean by wet and dry growing or what regions of the world that would cover. It would be clearer introducing FIGS as early as possible to clear up the confusion.
Line 45 -51: The term fava bean and “crop” is often used but it is not clear if these are interchangeable with fava bean or if “crop” is a more general use.
You could break up the paragraph by fava bean and other important crops to improve the structure
Line 55 – Define dry and wet regions of the world
Lin 58 – This is an example of structure how you could highlight the importance of your work and have a transition. Here instead you could write the following in your own words.
The fact is that breeders don’t breed directly for root traits traditionally and often focus on yield or resistance to biotic and abiotic stress. FIGS study is a great example of how the important underground structure of a crop is often overlooked, a major factor being the low-throughput and destructive nature of the process. To address this lack of knowledge of below ground contribution to drought stress adaptation, we incorporate high-throughput non-destructive methods.
86 -90 – Needs restructuring

I can’t go line for line throughout the paper but I recommend revisiting the paper and improving the structure and grammar.

Experimental design

2. Experimental Design
The experimental questions and design was well defined overall and had useful details so I could compare to previous similar studies.

There were a couple parts that I ask for improvement.

The germplasm was grown at separate times because of a limitation of space. Were there any temperature, humidity or other measurements taken over this timeline? What sort of variation was found between the experimental blocks? Where the blocks randomized within a growing interval or across the intervals. For example, was an accession represented within at each sowing time point?

Improve consistency of the terms such as days after sowing or days after treatment, but in some cases use terminology like “5th week” (Line 18) or “one week later (Line 188)

Planting and treatment management (portion)
Were there any rhizobacteria inoculation? Ex Rhizobium leguminosarum
The FIGS paper was referenced when choosing dry-adapted or wet-adapted. Maybe reference table 4 or outline major traits you based your selection for the 89 accessions.
Making the comparison of the automated acquisition of data with wash and preserved roots manually was a very useful comparison to estimate the accuracy of the high throughput technique.

Validity of the findings

3. Validity of the findings
The provided data analyses were acceptable, however, there were several opportunities to dig deeper into the data. I suggest reading some papers on high throughput phenotyping.
Dissecting the Phenotypic Components of Crop Plant Growth and Drought Responses Based on High-Throughput Image Analysis
The Plant Cell Dec 2014, 26 (12) 4636-4655; DOI: 10.1105/tpc.114.129601
A Versatile Phenotyping System and Analytics Platform Reveals Diverse Temporal Responses to Water Availability in Setaria
Fahlgren, Noah et al. Molecular Plant, Volume 8, Issue 10, 1520 - 1535

Not much detail on the handling of the data, such as a test for normal distribution before using Independent samples test. Any outliers removed? A number of samples equal among all samples?

Germplasm Survey
It was not clear if DAS or DAT were the values taken?? What day were the data collected?

Table 2: Not clear if significant difference were found between wet- and dry- accessions or an ANOVA was used to compare all three. Was the mean take of all replicates first, and how much variation was found within the replicates?

Line 209: Use numbers in the results. A negative correlation is too general of a description, especially when the negative correlation was -0.04

Table 3:
This should be a matrix with scatter plots and fit line

Line 214: Could use total dry weight by adding root and shoot dry weight

Root Phenotyping

Line 228: DS70622 had the longest 2nd order roots in well-watered treatment, so it is not surprising it has the smallest difference. The same table with percent change would tell us what plant had the smallest change regardless of size in well-watered. How do you determine which number is more useful for drought resistance or avoidance?

Line 236: Were these numbers for all plants, only well-watered or less-watered

Line 238: At the end of the treatment period for all treatments? (or less or well-watered)

Line 248 – 251: How does the author face the difference between well-watered and water-limited lines represented in the rhizotron? Differences between visible roots from each treatment could be significant.

Additional data analysis

Having multiple traits, the author should consider using principle component analysis and other multivariate methods to revisit all of the data. There is discussion of the time series of the experiment but no figures with days and trait measurements.

Results
Concentrated on outliers as sources for future studies. The figures don’t visually depict the outliers well. The author lists some outliers as max. or min., but it’s too difficult for the reader to make their own comparisons. I suggest ordering accessions by rank either with transformed data or raw data.
Line 279: “The outliers above regression line (Fig. 1)….” This is a great example of when this would be useful, however there is no regression line in the figure.
Overall structure of Results needs work, for example;
Line 272: The first sentence mentions chlorophyll concentration, but discusses root mass fraction in other papers to back up the possibility that RMF as potential source for breeding.
What is more important RMF at well-watered and dry conditions? Less of a root mass reduction in limited-water or percent change of root mass from well-watered to less-watered? It’s not clear why RMF is the major trait focused on except for the fact it was an outlier. More citations of the physiological uses of having higher root mass fraction would help the case. Citations of other correlations in useful, but citations of physiological connection is more powerful similar to Line 294. However, this acknowledge deeps roots is a benefit, not so much root mass overall.

Additional comments

There is no doubt that roots are far behind in the field of plant phenotyping and this paper will add useful results for an important crop combating common issues such as water stress. This paper uses previously defined groups of fava bean with a new high-throughput, non-destructive technique and identifies useful traits for further breeding.
Overall, I think there should be some major changes in the paper which I hope will improve the overall quality of the paper to its highest potential before publishing.

---

## Round 0.2 · Minor Revisions

Your revised manuscript has been seen by the original reviewers. Both reviewers felt the revision addressed their previous technical concerns. The additional screening requested by reviewer 1 would be useful, but is beyond the scope of this manuscript. Reviewer 2 had a few minor comments that should be addressed before this manuscript is published.

·

Basic reporting

no comments

Experimental design

no comments

Validity of the findings

no comments

Additional comments

I would like to thank the authors for the improvements they have made for the manuscript. I think that is a good idea to look at these results as a root diversity survey instead of direct use in breeding. I still believe the major limitation for this experiment is the lack of germplasm survey (88 genotypes) in water-deficit condition. Stomatal conductance, canopy temperature, chlorophyll concentration and shoot dry weights were measured at well-watered condition. These traits in water-available condition might be independent of the root architecture. Thus using these criteria to choose a subset of potential drought tolerant/susceptible genotypes for root phenotyping is not a very convincing approach. I highly encourage the authors to screen the 88 genotypes in the water-deficit conditions and then choose the drought tolerant/susceptible genotypes based on those results. I highly appreciate the author’s efforts to shed lights on the root architecture, as this is one of the bottlenecks in agricultural researches.

Reviewer 2 ·

Basic reporting

Overall, this manuscript is very clear in its all subjects under the basic reporting requirements. I have only a few suggestions below:


Line 53: For example, line…. (example of experiment)
Line 193: Could put all traits and descriptions in table format
Line 205: example images in supplementary may be useful (never mind for the rhizo images but could include scan images)
Line 242: Sentence is not clear at first, maybe introduce the concept of a, t x a
Line 245: What percentage on average and did it change for each treatment? Could contribute fewer resources to laterals. You have said you did check this and didn’t find a change, but no change is also interesting

Experimental design

Meets the standards of the aims and scopes for this journal and as a reviewer. The question and hypotheses were well defined and the methods of the project were spelled out in detail and easy to follow their logical approach to test their hypotheses.

Validity of the findings

no comment

Additional comments

The improvements to the paper met and sometimes exceeded expectations.

---

## Round 0.3 · accepted · Accept

Your revised manuscript has addressed the previous concerns raised by the reviewers.